# Identifying and describing alcohol-related paediatric emergency department attendances amongst under 16 year olds including time trends, incidence rates, and sociodemographic factors associated with alcohol-related harm

Nicholas Davies[1¤]*, Dimitrios Charalampopoulos[2], Abigail Rose[3], Shrouk Messahel[4]

1 Department of Psychology, Institute of Population Health, University of Liverpool, Liverpool, Merseyside, United Kingdom, 2 Public Health, Policy & Systems, Institute of Population Health, University of Liverpool, Merseyside, United Kingdom, 3 School of Psychology, Institute of Health Research, Liverpool John Moores University, Liverpool, Merseyside, United Kingdom, 4 Alder Hey Children's Hospital Trust, Liverpool, Merseyside, United Kingdom

¤ Current Address: School of Public and Allied Health, Liverpool John Moores University, Liverpool, Merseyside, United Kingdom
* Nicholas.davies@liverpool.ac.uk

## Abstract

This study used ED attendance data to identify and describe alcohol-related attendances and identify sociodemographic factors associated with risk of alcohol-related harm. Between 2011 and 2022, N = 774 patients had at least one alcohol-related attendance. Descriptive statistics were computed to show the sociodemographic profile of patients, and the number of attendances were aggregated by calendar year. Attendance rates per 100,000 were computed using UK national mid-year population estimates. Associations between sociodemographic characteristics and severity-related outcomes (including whether the patient was admitted to hospital, and whether the patient had multiple attendances) and patient safeguarding concerns were explored through logistic regression. Most attendances were female (73.90%) and around a quarter (26.75%) were admitted to hospital during at least one attendance. Attendances per year and percentage resulting in admission decreased during the study period. These changes were reflected by attendance rates which fell for boys (50.65 to 15.25 per 100,00) and girls (94.83 to 36.90 per 100,000) between 2011 and 2022. Females were less likely than males to be admitted to hospital (OR=0.61 [95%CI = 0.43, 0.88], p = .008). Additionally, younger adolescents were more likely to have reported safeguarding concerns (OR=0.26 [95%CI = 0.12, 0.56], p = .001). Finally, patients with safeguarding concerns and non-White ethnicity was associated with multiple attendances during the study period (OR=3.84 [95%CI = 1.74, 8.44], p = .001; OR=3.25 [95%CI = 1.14, 9.28], p = 0.28). Alcohol-related attendances have declined over the last 11 years, and harms may differ between

**Data availability statement:** The descriptive data underlying this research is provided in the manuscript and supporting information. The individual-level data underlying this study cannot be made publicly available due to restrictions outlined in the Data Management Plan agreed with the NHS provider (Alder Hey Children's Hospital NHS Trust, who provide care to children and young people). The dataset contains pseudo-anonymised individual-level information, and sharing it openly would pose a risk to participant confidentiality, even with anonymisation safeguards in place. These restrictions comply with NHS data governance policies and ethical requirements. Contact information: Alder Hey Children's NHS Foundation Trust Eaton Road Liverpool, L12 2AP Registration Number GB654938985 Email: alder.centre@alderhey.nhs.uk Telephone: 0151 252 5391.

**Funding:** This research was part of a PhD studentship funded by an external sponsor (Hugh Greenwood Legacy, Alder Hey Children's Hospital) and the University of Liverpool. The studentship was awarded to AR and SM and the PhD student was ND. The funders had no involvement in the methods, analysis, or interpretation of the research. Hugh Greenwood (https://liverpoolhealthpartners.org.uk/the-hugh-greenwood-legacy-for-childrens-health-research/) University of Liverpool (https://www.liverpool.ac.uk/).

**Competing interests:** The authors have declared that no competing interests exist.

genders. While most attendances were female, males were more likely to be admitted to hospital. Particular subgroups may be vulnerable to harms, including younger and non-White patients, potentially illustrating that adverse childhood experiences and marginalised characteristics are associated with harmful alcohol consumption. This study confirms that admissions data is likely to underestimate the true numbers of young people experiencing harm, and future research should conduct more robust analyses using attendance data to further understand risk factors.

## Introduction

Trends in alcohol consumption amongst under 18 year olds have changed over the past 20 years, with multiple datasets suggesting a reduction in underage alcohol use [1]. There are concerns, however, that polarisation effects may occur whereby those engaging in the most harmful patterns of alcohol consumption do not reduce their drinking despite overall declines in the general population [2]. In addition, population level changes in the prevalence of underage drinking raise questions around whether the sociodemographic risk factors associated with harmful alcohol use have also changed or remained the same.

Estimating the rates of underage alcohol consumption mainly relies on national school-based survey data, such as the Smoking Drinking and Drug use Among Young People (SDD) survey in England. The SDD shows that between 2003 and 2014, the percentage of 11–15 year olds in England who reported ever having consumed an alcoholic drink fell from around 60% to 40% [3]. A slight change to the wording of the alcohol question means it is not possible to directly compare pre-2016 data to newer data, although, the prevalence of ever having consumed alcohol fell again in the most recent dataset (from 40% to 37%) which could suggest alcohol use is still decreasing [4]. While national survey data is useful for tracking trends in the prevalence of self-reported alcohol use, the survey may not be as suited to capturing information around the prevalence of alcohol-related harm. For instance, the SDD questions mainly focus on the prevalence, quantity, and frequency of consumption rather than consequences of alcohol use (e.g. harms). In addition, such surveys are delivered in schools and might not capture higher risk individuals such as young people who are absent from school on the day of the survey. This means survey data is often supplemented with hospital admissions data to provide a more complete view around the prevalence of underage alcohol use and harms.

Hospital data provides a valuable lens for examining alcohol-related harms, as individuals engaging in riskier patterns of alcohol use may require hospitalisation for alcohol-related reasons such as acute intoxication or alcohol-related injuries [5]. An indication of the prevalence of alcohol-related hospital admissions in England is available through the Local Alcohol Profiles for England (LAPE), which are routinely published by the Office for Health Improvement and Disparities (OHID). This distinguishes between alcohol-related and alcohol-specific admissions. Alcohol-related admissions capture a broader range of admissions where alcohol is either a primary

or one of the secondary diagnoses. Alcohol-specific admissions, however, only capture admissions where alcohol is the primary diagnosis. As a result, admissions for alcohol-related conditions (for all age groups) were higher (2367 per 100,000) than for alcohol-specific conditions (664.1 per 100,000) between 2018 and 2020 [6]. Estimates, however, include all ages and are not specific to under 18 year olds.

The LAPE data only includes one indicator specific to under 18 year olds, which is the narrower measure of admission episodes for alcohol-specific conditions. The latest figures show that between 2018/2019 and 2020/2021, there were 29.3 alcohol-specific admissions per 100,000 population under 18 year old [6]. A report in 2016 compared hospital admissions data with national survey data, and found that the decline in the prevalence in underage drinking has occurred alongside concurrent declines in alcohol-specific hospital admissions [7]. There are, however, some limitations regarding the use of hospital admissions data to estimate the prevalence of underage alcohol-related harm. Firstly, analyses which rely on the narrower measure of alcohol-specific admissions are likely to underestimate the number of admissions where alcohol was a factor in the diagnosis even if not listed as the primary diagnosis. This can be said, for example, if a person was admitted to hospital with an injury (e.g. broken leg) as the primary diagnosis, however, the injury was sustained while the person was intoxicated with alcohol (i.e. intoxication was a secondary diagnosis). This is an issue particularly when studying alcohol for under 18 year olds, as the LAPE statistics only include data for alcohol-specific admissions, whereas for adults there are multiple measures including the broader alcohol-related admissions indicator.

In addition to this, alcohol-specific or alcohol-related admissions data do not show the numbers of people presenting to ED departments with alcohol-related issues. This can be said as a person may have attended the ED for an alcohol-related reason, however, they may have been treated and discharged within the ED rather than admitted to hospital. There is a difference, therefore, between hospital attendance data and hospital admissions data, with the former more difficult to obtain and less well understood. For instance, research across several EDs in the North West of England found that for each alcohol-related admission for under 18 year olds, there were multiple alcohol-related attendances which were not captured in the admissions data [8].

Understanding the prevalence of underage alcohol-related harm requires examining not only overall figures but also key sociodemographic risk factors. Firstly, there are clear sex differences; for boys, alcohol-specific hospital admissions increase with age in a linear trajectory, whereas, for girls admissions increase sharply and peak around 15 years old [7]. This is important, as under 18 year olds are the only age group where girls show higher rates of alcohol-specific hospital admission compared to boys. This could suggest either young girls are at higher risk of alcohol-related harm, or there is a perception of greater harm for females who consume alcohol resulting in them attending EDs. This also suggests the sociodemographic risk factors associated with alcohol-related harm might differ between under 18 year olds and adults. This is also supported by national survey data, which shows that adolescent girls aged 11–15 years old are more likely than boys to report intoxication during the past month [9].

In addition, adolescents of White ethnicity and from higher socioeconomic status are more likely to report alcohol consumption than individuals from other ethnic groups or those from lower socioeconomic positions [9]. There is also some evidence of the Alcohol Harms Paradox for underage alcohol consumption, whereby individuals from lower socioeconomic backgrounds experience greater risk of alcohol-related harms despite consuming less alcohol compared to those from higher socioeconomic backgrounds [7].

Finally, previously established predictors of alcohol-related harm appear to follow different patterns in more recent cohorts of young people. For example, mental health difficulties are a well-established predictor of alcohol-related harm among adults, showing associations with heavier alcohol consumption [10] and alcohol use disorder diagnosis [11]. However, while alcohol consumption overall has declined among under 18 year olds, mental health difficulties have either increased or remained stable. For example, in England the rates of having a probable mental disorder increased from 12.1% to 16.17% for 7–16 year olds between 2017 and 2020 [12]. This could indicate that previously established risk factors for alcohol-related difficulties are changing for the most recent cohort of under 18 year olds, with the association

between mental health and alcohol consumption becoming less clear. Cohort analysis, however, shows some evidence that alcohol and mental health difficulties show stronger associations in more recent birth cohorts, despite a reduction in general prevalence of alcohol and substance use [13].

The limitations of relying on school-based survey data and hospital admissions data to describe the prevalence of alcohol-related harm among under 18 year olds suggests research using hospital attendance data is required to better understand this issue. In addition, the changing trends in the prevalence of underage alcohol use raise questions around how this has affected the numbers of under 18 year olds presenting to EDs with alcohol-related problems. Finally, the differences in sociodemographic risk factors for alcohol-related harms in under 18 year olds compared to adults show that a better understanding is needed to describe the sociodemographic characteristics associated with alcohol-related ED attendances in younger drinkers specifically. This is especially important given the alcohol harm paradox; if those from lower socioeconomic groups make up most alcohol-related hospital attendances, this suggests there may be particular groups who continue to be more susceptible to harm despite a population level decline in young people's alcohol use which may suggest widening health inequalities.

Using data from a paediatric ED in the North West of England between 2011 and 2022, the current research has three aims: (1) to describe the sociodemographic profile of 5–15 year olds with at least one alcohol-related attendance, (2) to describe temporal trends of alcohol-related attendances, and (3) to examine the association between sociodemographic risk factors, safeguarding concerns, and severity-related outcomes such as whether the patient was admitted to hospital and whether the patient had multiple alcohol-related attendances.

## Materials and methods

### Data set

Data on hospital attendances between 1/1/2011 to 31/12/2022 were extracted from Meditech, an electronic database containing information on ED attendances at a paediatric ED in the North West of England. The data were accessed on 23rd November 2023. To identify alcohol-related visits, we analysed two variables: the visit reason, a free-text field detailing the nature of the attendance (e.g., "alcohol intoxication"), and the sub-diagnosis code, a standardised set of codes outlining the diagnosis condition [14]. Attendances were classified as alcohol-related if the visit reason included the term "alcohol" and/or if the sub-diagnosis code was marked as "alcohol" or "alcohol intoxication". We also applied an age criterion, excluding patients under 5 – as their attendance typically does not involve voluntary alcohol consumption, i.e. accidental ingestion of alcohol gel – and those 16 and older, as the setting is a paediatric hospital with scant cases beyond this age, skewing representativeness. In addition, 8 duplicate records were identified by matching arrival dates and patient IDs and were excluded.

### Data sharing agreement

The study was approved by HRA and Health Care Research Wales (HCRW) on 6th May 2023 (IRAS Project ID 212772; REC Reference 23/HRA/1065). A data sharing agreement between the University of Liverpool and Alder Hey Children's NHS Trust was signed on the 7th September 2023. Written/oral consent was not required as study was a secondary analysis with pseudonymised clinical data. Individual level data were not made publicly available due to restrictions outlined in the Data Management Plan agreed with the NHS provider. Data sharing, therefore, was not possible as the authors had access to information that could identify individual participants during and after data collection.

### Variables

**Demographic variables.** Each patient was given a patient ID number to differentiate between patients with more than one alcohol-related attendance during the study period. Patient sex was coded as either male or female, from the patient's sex at birth. Age in years during each attendance was computed from date of birth. To avoid issues with repeated

observations, age at first attendance (in years) was used for the logistic regression models to account for patients with multiple attendances during the study period.

Age was coded as a categorical variable (12 years or under, 13 years old, 14 years old, 15 years old).

Ethnicity was recoded from the broader NHS codes for ethnicity [15] into 3-level categorical variable (White, any other ethnic groups, or not stated/unknown).

Reported safeguarding concerns was computed as a binary variable for each attendance. Such concerns included reasons such as children in social care, previous referral to mental health services or reports of domestic violence. The variable indicated whether each patient has ever had an alcohol-related attendance with reported safeguarding concerns or not.

Index of Multiple Deprivation (IMD) quintile was computed through merging patient postcode data with national English indices of deprivation 2019 data [16]. This was recoded into quintiles: a 1–5 score indicating relative deprivation, with 1 indicating the most deprived and 5 being least deprived.

**Temporal variables.** A datetime variable was computed indicated the date and time of each attendance. In addition, a datetime variable was computed showing the date and time in which the patient was admitted to hospital (for patients admitted to hospital following an alcohol-related attendance).

**Severity-related variables.** Total alcohol-related attendances was computed for each patient, showing the total number of alcohol-related attendances each patient had during the study period. This was recoded as a binary variable for analyses (single versus multiple alcohol-related attendances).

Admission was coded as a binary variable to indicate whether each patient had ever been admitted to hospital following an alcohol-related attendance.

## Data analyses

All data were analysed using Stata 18 [17]. Statistical significance was set at the $p < .05$ level for all statistical tests. Descriptive statistics were computed with individual level data (i.e. one observation per patient); for patients with multiple alcohol-related attendances only the first observation was kept. Sociodemographic variables were tabulated followed by severity-related variables. Tables showed the total and percentages within each level of the variables, and the totals between sex were compared with Chi Square tests.

Incident cases of alcohol-related attendances between 2011 and 2022 were described by aggregating the data by calendar year. Aggregated data was graphed showing the total number of attendances per calendar year in a bar chart. A stacked bar chart was used to show the difference in totals between males and females each year, and the percentages of attendances which resulted in admission to hospital.

The annual rates of alcohol-related attendances were determined by dividing the number of attendances each year by the population of 5 to 15-year-olds in Merseyside, using mid-year estimates from the Office for National Statistics (ONS) [18]. While ONS denominators may not precisely reflect the catchment population, they provide a stable basis for trend analysis over time, accounting for demographic changes. Alcohol-related attendance rates were computed separately for each gender to facilitate comparative analysis. 95% confidence intervals were also calculated for each rate using the Wilson Score method [19] as recommended by the Association of Public Health Observatories [20]. Attendance rates between 2011 and 2022 were presented in a bar chart (with 95%CI displayed as error bars), with separate bars for males and females to allow gender comparisons.

The relationship between sociodemographic factors, safeguarding concerns, and the severity of alcohol-related incidents, such as hospital admissions and multiple attendances, was examined using maximum likelihood logistic regression models. The association between sociodemographic variables and reporting of safeguarding concerns was also explored through logistic regression. For each covariate, both unadjusted and adjusted odds ratios were calculated, with the latter accounting for all additional variables in the model. Corresponding 95% confidence intervals and p values were

also reported alongside each odds ratio. Due to small cell counts, IMD quintile was treated as a continuous variable in the analysis. There was a small amount of missing data for IMD (n = 2) and ethnicity (n = 51), therefore a complete case analysis was conducted.

## Results

### Descriptive statistics

Table 1 shows that alcohol-related attendances were more common for females (73.90%) than males. Attendances increase with age, although the percentages at different ages differ between males and females, including a higher proportion of females at ages 12–13 and higher proportion of males at 14–15. The majority of attendances were of White ethnicity (89.41%), and around a quarter of patients (26.74%) were admitted to hospital. According to the IMD scores, the majority of patients (70.34%) were in the lowest 20% of deprivation. Most patients had a single alcohol-related attendance (95.99%, M = 1.05, SD = 0.26) and the maximum number of alcohol-related attendances was four (Table 2). Most patients (84.37%) did not present with safeguarding concerns, however, percentage with safeguarding concerns appeared higher for females compared to males.

### Alcohol-related attendances and admissions by year

The first aim was to explore incidence of alcohol-related attendances and admissions over time. Fig 1 summarizes the number of alcohol-related attendances per calendar year, by gender. Fig 1 also shows the percentage of attendances which resulted in admission to hospital.

**Table 1. Demographic profile of children under the age of 16 (n = 774) with one or more alcohol-related attendance between 2011 and 2022, by sex.**

| | Full sample n = 774 n (%) | Males n = 202 (26.10%) n (%) | Females n = 572 (73.90%) n (%) | p (Males versus females) |
|---|---|---|---|---|
| Age at first attendance (years) | | | | <.001** |
| ≤12 | 43 (5.56) | 12 (5.94) | 31 (5.42) | |
| 13 | 135 (17.44) | 25 (12.38) | 110 (19.23) | |
| 14 | 280 (36.18) | 57 (28.22) | 223 (38.99) | |
| 15 | 316 (40.83) | 108 (53.37) | 208 (36.36) | |
| Ethnicity | | | | .031* |
| White | 692 (89.41) | 185 (91.58) | 507 (88.64) | |
| Any other ethnic group | 40 (5.17) | 13 (6.44) | 27 (4.72) | |
| Not stated or unknown | 42 (5.43) | 4 (1.98) | 38 (6.64) | |
| IMD | | | | .046* |
| 1 | 543 (70.34) | 130 (65.00) | 413 (72.20) | |
| 2 | 86 (11.14) | 23 (11.50) | 63 (11.01) | |
| 3 | 69 (8.94) | 22 (11.00) | 47 (8.22) | |
| 4 | 60 (7.77) | 17 (8.50) | 43 (7.52) | |
| 5 | 14 (1.81) | 8 (4.00) | 6 (1.05) | |
| Safeguarding concerns | | | | .017* |
| No | 653 (84.37) | 181 (89.60) | 472 (82.52) | |
| Yes | 121 (15.63) | 21 (10.40) | 100 (17.48) | |

*p<.05, **p<.001 (Chi Square test), n=2 missing values for IMD.

**Table 2. Severity-related outcomes for male and female children with at least one alcohol-related attendance between 2011 and 2022 (n = 774).**

| | Full sample n = 774 n (%) | Males n = 202 (26.10%) n (%) | Females n = 572 (73.90%) n (%) | p (Males versus females) |
|---|---|---|---|---|
| Attendances (n) | | | | .970 |
| 1 | 743 (95.99) | 194 (96.04) | 549 (95.98) | |
| >1 | 31 (4.01) | 8 (3.96) | 23 (4.02) | |
| Ever admitted to hospital | | | | .027* |
| No | 567 (73.26) | 136 (67.33) | 431 (75.35) | |
| Yes | 207 (26.74) | 66 (32.67) | 141 (24.65) | |

*p < .01 (Chi Square test).*

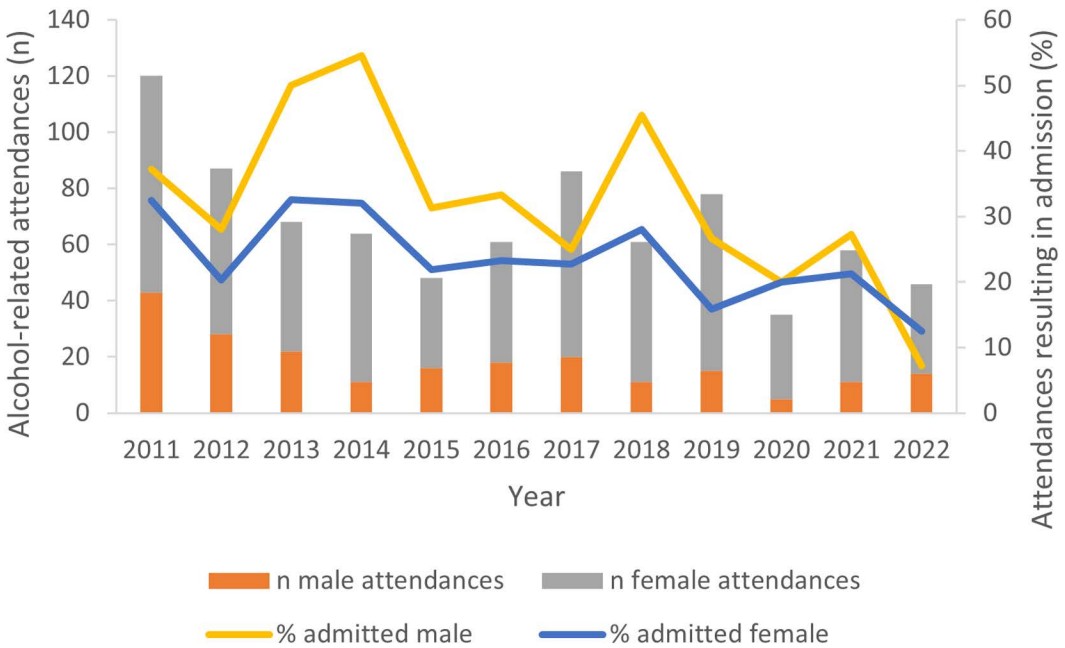

**Fig 1. Alcohol-related attendances (n) and percentage of attendances resulting in admission between 2011 and 2022 (n = 812) for males and females.**

Fig 1 shows that the number of alcohol-related attendances per year appear to have decreased between 2011 and 2022, however, at each time point there were more female than male patients. The percentage change in attendances between 2011 and 2022 for males was an 80% reduction, compared to a 75% reduction for females. The percentage of attendances resulting in hospital admission also decreased, from 37.21% to 7.14% for males and from 32.47% to 12.50% for females. A higher percentage of male attendances resulted in admission to hospital than female attendances, although, the difference between these appears to have reduced over time. See S1 Table for exact data relating to Fig 1.

Fig 2 shows that attendance rates for female attendances compared to male attendances are higher each year using population estimates for 5- to 15-year-olds in Merseyside. This suggests that relative the local population, a higher proportion of alcohol-related attendances are female compared to male. The figure also shows that the proportion of alcohol-related attendances compared to the local population appears to have decreased for both males and females during the study period. Attendance rates per 100,000 for boys fell from 50.65 (95%CI = 37.61, 68.22) in 2011 to 15.25

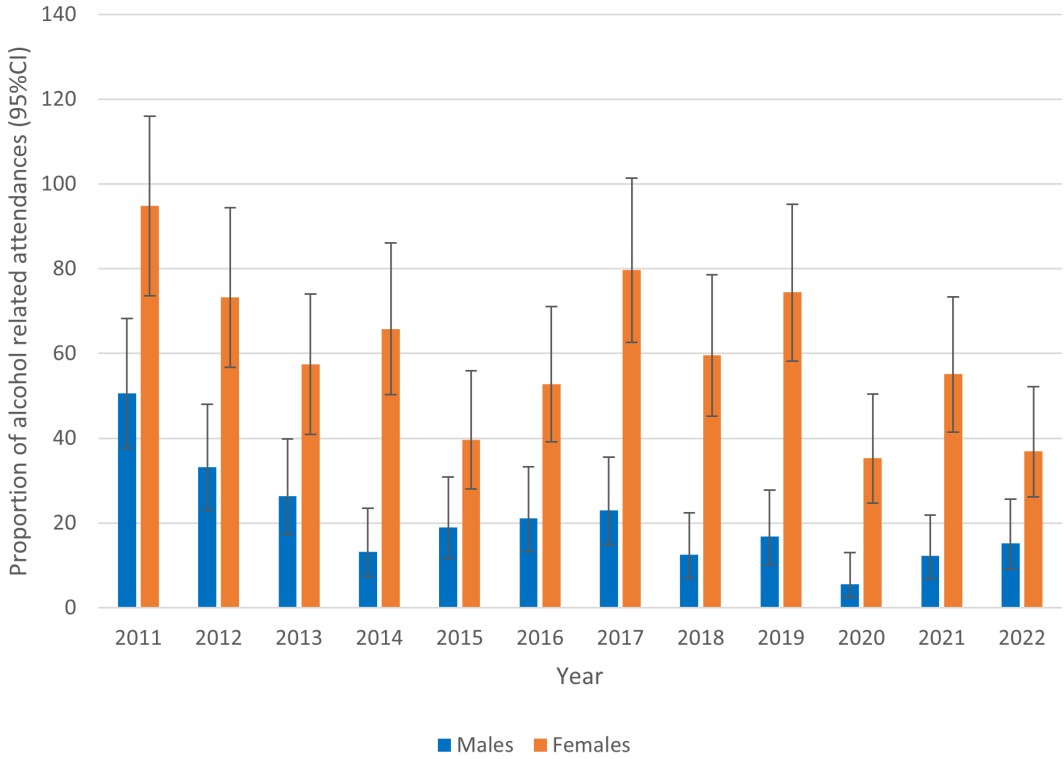

**Fig 2. Annual alcohol-related attendance rates per 100,000 using mid-year population estimates for 5- to 15-year-olds in Merseyside (by gender).**

(95%CI = 9.09, 25.61) in 2022. Similarly, for girls, attendances rates fell from 94.83 (95%CI = 73.66, 116.00) in 2011 to 36.90 (95%CI = 26.14, 52.09) in 2022 (see S2 Table). This suggests that the reduced alcohol-related attendances per year likely reflect genuine reductions, rather than being influenced by local demographic changes.

**Association between sociodemographic characteristics and severity of attendance**

Table 3 shows the association between sociodemographic variables and being admitted to hospital for children with at least one alcohol-related attendance between 2011 and 2022. Females were less likely than males to be admitted to hospital following an alcohol-related attendance (Adjusted OR=0.61 [95%CI = 0.43, 0.88], p = .008). No significant associations were found with age, ethnicity, or IMD.

Table 4 shows the association between sociodemographic characteristics and having multiple alcohol-related attendances amongst children with at least one alcohol-related attendance between 2011 and 2022. Children from any other ethnic background were more likely than White patients to have recurrent alcohol-related attendances, even after adjusting for sociodemographic variables (Adjusted OR=3.25 [95%CI = 1.14, 9.28], p = 0.28).

In addition, children presenting with reported safeguarding concerns at any visit were more likely to have multiple alcohol-related attendances after adjusting for sociodemographic differences (Adjusted OR=3.84 [95%CI = 1.74, 8.44], p = .001). This suggests that non-White children and those with safeguarding concerns are more likely to have multiple alcohol-related attendances amongst children with at least one alcohol-related attendance. Age, sex and IMD, however, were not significantly associated with having multiple alcohol-related attendances.

**Table 3.  Association between sociodemographic characteristics and hospital admission in 5- 15 year olds with one or more alcohol-related attendance between 2011 and 2022.**

| | Not admitted to hospital n (%) | Admitted to hospital n (%) | Odds ratio (95%CI) | p | Adjusted[1] Odds ratio (95%CI) | p |
|---|---|---|---|---|---|---|
| Sex | | | | | | |
| Male | 136 (67.33) | 66 (32.67) | 1.00 | | 1.00 | |
| Female | 431 (75.35) | 141 (24.65) | 0.67 (0.47, 0.96) | .027* | 0.61 (0.43, 0.88) | .008** |
| Age at first attendance | | | | | | |
| ≤12 | 29 (67.44) | 14 (32.56) | 1.00 | | 1.00 | |
| 13 | 100 (74.07) | 35 (25.93) | 0.73 (0.34, 1.53) | .398 | 0.75 (0.35, 1.59) | .450 |
| 14 | 201 (71.79) | 79 (28.21) | 0.81 (0.41, 1.62) | .559 | 0.85 (0.42, 1.71) | .654 |
| 15 | 237 (75.00) | 79 (25.00) | 0.69 (0.35, 1.37) | .291 | 0.69 (0.34, 1.38) | .290 |
| Ethnicity | | | | | | |
| White (Any) | 513 (74.13) | 179 (25.87) | 1.00 | | 1.00 | |
| Any other ethnic groups | 28 (70.00) | 12 (30.00) | 1.23 (0.61, 2.47) | .563 | 1.16 (0.57, 2.35) | .678 |
| Not stated or unknown | 26 (61.90) | 16 (38.10) | 1.76 (0.92, 3.36) | .085 | 1.87 (0.97, 3.61) | .060 |
| IMD quintile[2] | 1.61 (1.05) | 1.56 (1.05) | 0.96 (0.82, 1.12) | .570 | 0.94 (0.80, 1.10) | .546 |
| Safeguarding concerns | | | | | | |
| No | 482 (73.81) | 171 (26.19) | 1.00 | | 1.00 | |
| Yes | 85 (70.25) | 36 (29.75) | 1.19 (0.78, 1.83) | .416 | 1.15 (0.74, 1.78) | .546 |

*p<.05, [1]Adjusted for other predictors in the model [2]Continuous outcome (mean and SD).

**Table 4.  Association between sociodemographic characteristics and multiple alcohol-related attendances in under 16-year-olds with one or more alcohol-related attendance between 2011 and 2022.**

| | Single alcohol-related attendance n (%) | Multiple alcohol-related attendances n (%) | Odds ratio (95%CI) | p | Adjusted[1] Odds ratio (95%CI) | p |
|---|---|---|---|---|---|---|
| Sex | | | | | | |
| Male | 194 (96.04) | 8 (3.96) | 1.00 | | 1.00 | |
| Female | 549 (95.98) | 23 (4.02) | 1.02 (0.45, 2.31) | .970 | 0.98 (0.42, 2.30) | .959 |
| Age | | | | | | |
| ≤12 | 40 (93.02) | 3 (6.98) | 1.00 | | 1.00 | |
| 13 | 131 (97.04) | 4 (2.96) | 0.41 (0.09, 1.90) | .252 | 0.54 (0.11, 2.66) | .451 |
| 14 | 268 (95.71) | 12 (4.29) | 0.60 (0.16, 2.21) | .440 | 0.74 (0.19, 2.87) | .663 |
| 15 | 304 (96.20) | 12 (3.80) | 0.53 (0.14, 1.95) | .336 | 0.82 (0.21, 3.28) | .782 |
| Ethnicity | | | | | | |
| White (any) | 666 (96.24) | 26 (3.76) | 1.00 | | 1.00 | |
| Any other ethnicity | 35 (87.50) | 5 (12.50) | 3.66 (1.33, 10.10) | .012* | 3.25 (1.14, 9.28) | .028* |
| Not stated or unknown | 42 (100.00) | 0 (0) | | | | |
| IMD quintile[2] | 1.60 (1.05) | 1.58 (1.15) | 0.99 (0.70, 1.39) | .934 | 0.98 (0.69, 1.38) | .897 |
| Safeguarding concerns | | | | | | |
| No | 634 (97.09) | 19 (2.91) | 1.00 | | 1.00 | |
| Yes | 109 (90.08) | 12 (9.92) | 3.67 (1.73, 7.78) | .001** | 3.84 (1.74, 8.44) | .001** |

*p<.05,**p<.01, [1]Adjusted for other predictors in the model [2]Continuous outcome (mean and SD).

## Association between sociodemographic characteristics and reported safeguarding concerns

Table 5 shows the association between sociodemographic characteristics and reported safeguarding concerns amongst children with at least one alcohol-related attendance between 2011 and 2022. Female gender was positively associated with presence of safeguarding concerns (OR=1.83 [95%CI=1.11, 3.01], p=.018), however, this association was not significant after adjusting for sociodemographic variables. In addition, those aged 15 years old were less likely than those aged 12 or less to have reported safeguarding concerns, both before and after adjusting for sociodemographic covariates (Adjusted OR=0.26 [95%CI=0.12, 0.56], p=.001). This suggests that older adolescents (aged 15) are less likely than younger adolescents to have reported safeguarding concerns.

## Discussion

The current research identified alcohol-related attendances at a paediatric ED, describing the number of incident cases per year, and explored the associations between sociodemographic characteristics and severity-related outcomes. Alcohol-related attendances appeared to decrease during the 11-year study period, and the sociodemographic profile of these attendances showed similarities with previously established characteristics in terms of age, gender, ethnicity and socioeconomic status. There were gender differences, with more girls having an alcohol-related ED attendance, but males more likely than females to be admitted to hospital as a result of alcohol. Younger adolescents were more likely to have reported safeguarding concerns, and those with reported safeguarding concerns were more likely to have multiple alcohol-related attendances during the study period. There was also an association between non-White ethnicity and multiple attendances, suggesting that non-White patients with an alcohol-related attendance were more likely than White patients with an alcohol-related attendance to have multiple attendances.

The sociodemographic profile of the attendances showed similarities with previously established sociodemographic risk factors for alcohol-related harm amongst young people. This included a higher proportion of female patients, older patients, patients of White ethnicity, and from the most deprived regions as indicated by IMD scores. Similar sociodemographic patterns have been demonstrated in national school-based surveys on underage alcohol consumption [4,9] and alcohol-specific hospital admissions data [7]. In addition, the decline in the number of alcohol-related attendances during

**Table 5. Association between sociodemographic characteristics and reported safeguarding concerns in under 16 year olds with one or more alcohol-related attendance between 2011 and 2022.**

| | No safeguarding concerns n (%) | Safeguarding concerns n (%) | Odds ratio (95%CI) | p | Adjusted[1] Odds ratio (95%CI) | P |
|---|---|---|---|---|---|---|
| Sex | | | | | | |
| Male | 181 (89.60) | 21 (10.40) | 1.00 | | 1.00 | |
| Female | 472 (82.52) | 100 (17.48) | 1.83 (1.11, 3.01) | .018* | 1.53 (0.91, 2.56) | .108 |
| Age | | | | | | |
| ≤12 | 31 (72.09) | 12 (27.91) | 1.00 | | 1.00 | |
| 13 | 108 (80.00) | 27 (20.00) | 0.65 (0.29, 1.42) | | 0.61 (0.28, 1.36) | .229 |
| 14 | 226 (80.71) | 54 (19.29) | 0.62 (0.30, 1.28) | | 0.60 (0.29, 1.25) | .171 |
| 15 | 288 (91.14) | 28 (8.86) | 0.25 (0.11, 0.54) | <.001*** | 0.26 (0.12, 0.56) | .001** |
| Ethnicity | | | | | | |
| White (any) | 592 (85.55) | 100 (14.45) | 1.00 | | 1.00 | |
| Any other ethnicity | 31 (77.50) | 9 (22.50) | 1.72 (0.79, 3.72) | .169 | 1.69 (0.76, 3.73) | .196 |
| Not stated or unknown | 30 (71.43) | 12 (28.57) | 2.37 (1.17, 4.78) | .016* | 2.13 (1.04, 4.37) | .040* |
| IMD quintile[2] | 1.61 (1.04) | 1.53 (1.09) | 0.93 (0.76, 1.13) | .446 | 0.93 (0.76, 1.13) | .466 |

*p<.05, **p<.01, ***p<.001, [1]Adjusted for other predictors in the model [2]Continuous outcome (mean and SD).

the study period also fits with estimates of alcohol use in the general population, which have declined significantly for young people since the early 2000s [21]. Alcohol-specific hospital admissions have also declined in a similar way, consistent with the findings from the current research [7]. Overall, this is evidence that the decrease in population level alcohol consumption amongst young people may be translating into fewer alcohol-related attendances, which suggests reduced alcohol-related harm. The research also showed that most alcohol-related attendances identified during the study period did not result in hospital admission. This is consistent with previous research using hospital attendance data [8], which highlights that tracking alcohol-related harm through admissions (rather than ED attendance data) is likely to underestimate the true number of young people experiencing harm in secondary care settings.

The current study also found that total alcohol-related attendances and attendance rates were higher for females than males. Despite this, males were more likely than females to be admitted to hospital which could suggest that males experience more severe alcohol-related problems requiring hospitalisation. The more severe alcohol-related injuries for males is consistent with trends observed in adults; while gender differences in alcohol use are narrowing, males have historically consumed alcohol in higher quantities than females and experienced more alcohol-related harms [22]. The higher rates of alcohol-related attendances in females, however, could reflect differences between adults and adolescents in terms of drinking behaviour. Early adolescence is the only period where girls are more likely than boys to consume alcohol and report intoxication, whereas by older adolescence (i.e. age 17 and 18) boys overtake girls in terms of alcohol consumption [7].

Whether these differences translate into observable differences in alcohol-related hospital attendances or admissions, however, is unclear. A recent study, found that while males had a higher incidence of alcohol-related unintentional injuries than females, the incidence risk ratios between alcohol use and hospital contact for unintentional injuries were similar between genders, suggesting neither gender is at higher risk of alcohol-related harm [5]. Another study examining trends in alcohol-related injury admissions for adolescents also found evidence for gender differences in admission types [23]. Admission for violence-related injuries appeared more prevalent for males, however, for females self-harm related injuries were more common, and appeared to be increasing for some age groups [23]. Further research is needed to compare the nature of alcohol-related presentations in ED settings (e.g. violent injuries, unintentional injuries) between boys and girls, to understand how patterns of harm may differ between these groups. Studying these gender-related differences using attendance data is particularly important, as attendance data captures a wider range of incidents and is less well studied compared with admissions data.

Related to this is the observed higher rates of alcohol-related attendances in girls compared to boys. It is possible that this could reflect the impact of gendered stereotypes. For example, concern that female alcohol use may increase risk of sexual assault or violence [24–26]. These cultural beliefs and attitudes may result in a gendered "vulnerability narrative" in which a girl's alcohol use is believed to be associated with higher risk compared to male alcohol use. This, in turn, could increase the likelihood of intoxicated girls being taken into ED settings, even if their drinking levels or risks of alcohol-related harm are lower than that of boys.

The research also found that younger patients were more likely to have reported safeguarding concerns. These concerns included a range of adverse experiences, including sexual abuse, domestic violence, and being in social care. Research shows that earlier exposure to adverse childhood experiences increases the risk of negative outcomes such as psychological distress [27,28]. Importantly, in the current study, reported safeguarding concerns were associated with increased risk of having multiple alcohol-related attendances. This suggests that amongst patients with an alcohol-related attendance, those exposed to early childhood adversities may be more likely to have repeated harms due to alcohol. This could also suggest that the trajectory into harmful alcohol use is quicker in children experiencing early childhood adverse experiences. This finding supports the association between exposure to adverse childhood experiences and health risk behaviours, which includes heavy alcohol use [29].

There was also an association between ethnicity and multiple alcohol-related attendances. While there were very few non-White patients with alcohol-related attendances, non-White patients were more likely than White patients to have

multiple attendances during the study period. This finding could appear unexpected, as survey data in the UK shows that non-White adolescents consume less alcohol than White adolescents [4]. Furthermore, analyses of engagement in multiple health risk behaviours also show that White compared to non-White individuals appear more likely to engage in risky health-related behaviours [30]. Some research, however, shows that while those from ethnic minority backgrounds tend to drink less overall, there is higher risk drinking amongst certain groups such as Irish Travellers or men belonging to the Sikh religion [31]. A limitation of the current research was that due to the low numbers of patients from ethnic minority groups, broader categories of 'White', 'non-White', or 'unknown' were used for the analyses, masking potential variations within groups. Alcohol consumption and harm clearly varies within ethnic minority groups, and therefore, the broader group used for the analyses will have included a wider range of patients from various ethnic groups. There are also limitations in coding practices in healthcare data more generally, which specifically affect patients from minority ethnic groups. Issues include variation in ethnic categories between data sources, and less accurate recording for ethnic minority groups compared to white patients [32]. In the current research, this could be reflected by the similar numbers in patients coded as non-White and those coded as not stated or unknown. These findings reflect current calls from researchers for better coding practices amongst healthcare providers to address health inequalities for those from minority ethnic groups [33].

There are some limitations to consider in relation to this study. Importantly, the research identified incident cases of alcohol-related attendances, and while these appeared to be declining over time, they may have been affected by other factors not related to alcohol consumption, such as demographic changes in the local catchment area, or differences in coding practices over time. To address this, incidence rates per 100,000 were calculated using mid-year population estimates for the local population. This suggested that relative to age and gender changes in the local population, the rates of alcohol-related attendances were falling. There are also more general limitations to relying electronic health record data, such as potential diagnostic coding errors. Research within ED settings shows that there are multiple barriers to obtaining information around underage alcohol use in practice [8], which could suggest that in some cases alcohol-related presentations are misdiagnosed. In addition, COVID-19 restrictions were implemented between 2021 and 2022 which may have resulted in reduced alcohol consumption given that young people who spent time with others outside their household were more likely to consume alcohol during the COVID-19 pandemic [9]. For this reason, the findings that alcohol harms have decreased over time should be interpreted with caution. In addition, the findings may not be generalised to other regions with different patterns of deprivation or case-mix profiles. Another limitation of this research is that while the research attempted to capture a broad range of alcohol-related incidents, relying on ED data only may underestimate cases where the perpetrator, rather than hospitalised victim have been under the influence of alcohol [34]. A clear strength of this research, however, was the use of attendance data rather than using admissions data only. This research demonstrates that a small percentage of alcohol-related ED attendances result in admission to hospital, meaning analyses using admissions data are likely to underestimate the true number of alcohol-related incidents requiring medical treatment amongst young people.

## Conclusions

This research found that alcohol-related ED attendances amongst young people appear to follow a similar sociodemographic pattern to other estimates of alcohol-harm amongst young people, such as population surveys and admissions data. There was evidence of gender differences in severity outcomes, with males more likely to be admitted to hospital, despite attendance rates being higher for females. The association between safeguarding concerns and multiple admissions suggests that children with adverse childhood experiences could be more vulnerable to alcohol-related harms. Reducing adverse childhood experiences should be a priority which will likely have a positive impact on reducing alcohol harm in the short and long terms. Future research should also explore how gender affects the type of attendance, including whether the reasons for the attendance and the actual (or perceived) risk of harm differ between male and female attendances. Given these limitations, future research could consider combining attendance data from multiple sites around the UK, to confirm the trends observed using regional data from a single ED.

## Supporting information

**S1 Table. Number of alcohol-related attendances per year and percentage admitted to hospital, by gender.**
(DOCX)

**S2 Table. Proportion of alcohol-related attendances per year compared to mid-year population estimates for 5 to 15 year olds in Merseyside (by gender).**
(DOCX)

## Author contributions

**Conceptualization:** Nicholas Davies, Abigail Rose, Dimitrios Charalampopoulos.

**Data curation:** Nicholas Davies, Dimitrios Charalampopoulos.

**Formal analysis:** Nicholas Davies, Dimitrios Charalampopoulos.

**Funding acquisition:** Abigail Rose.

**Investigation:** Nicholas Davies, Dimitrios Charalampopoulos.

**Methodology:** Nicholas Davies, Dimitrios Charalampopoulos.

**Project administration:** Nicholas Davies, Abigail Rose, Dimitrios Charalampopoulos.

**Resources:** Nicholas Davies, Abigail Rose, Dimitrios Charalampopoulos.

**Software:** Nicholas Davies, Dimitrios Charalampopoulos.

**Supervision:** Abigail Rose, Dimitrios Charalampopoulos, Shrouk Messahel.

**Validation:** Nicholas Davies, Abigail Rose, Dimitrios Charalampopoulos.

**Visualization:** Nicholas Davies, Abigail Rose, Dimitrios Charalampopoulos.

**Writing – original draft:** Nicholas Davies.

**Writing – review & editing:** Nicholas Davies, Abigail Rose, Dimitrios Charalampopoulos, Shrouk Messahel.

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
