## [Decision Letter · Decision Letter 0]

13 May 2025

Dear Dr. Davies,

Thank you for submitting your manuscript to PLOS ONE. After careful consideration, we feel that it has merit but does not fully meet PLOS ONE’s publication criteria as it currently stands. Therefore, we invite you to submit a revised version of the manuscript that addresses the points raised during the review process.

We look forward to receiving your revised manuscript.

Kind regards,

Nik Hisamuddin Nik Ab. Rahman

Academic Editor

PLOS ONE

Journal Requirements:

“We thank the Hugh Greenwood Legacy and the University of Liverpool for supporting this research.”

“This research was part of a PhD studentship funded by an external

sponsor (Hugh Greenwood Legacy, Alder Children’s Hospital) and the University of

Liverpool. The studentship was awarded to AR and SM and the PhD student was ND. The funders had no involvement in the methods, analysis, or interpretation

of the research.

Hugh Greenwood (https://liverpoolhealthpartners.org.uk/the-hugh-greenwood-legacy-for-childrens-health-research/)

University of Liverpool (https://www.liverpool.ac.uk/ )”

Reviewers' comments:

Reviewer's Responses to Questions

**Comments to the Author**

1. Is the manuscript technically sound, and do the data support the conclusions?

Reviewer #1: Yes

Reviewer #2: Yes

2. Has the statistical analysis been performed appropriately and rigorously?

Reviewer #1: Yes

Reviewer #2: Yes

3. Have the authors made all data underlying the findings in their manuscript fully available?

Reviewer #1: Yes

Reviewer #2: Yes

4. Is the manuscript presented in an intelligible fashion and written in standard English?

Reviewer #1: Yes

Reviewer #2: Yes

Reviewer #1: Overall, I think this article is on an important topic to look at alcohol-related paediatric ED attendances among patients under 16 years of age and provides valuable information on trends and associated harms. It presents new insights on the area and is well written. My general comments are:

Page 2, line 29: ONS abbreviation needs to be explained as not previously used or explained.

Abstract: Page 2-3: needs to include details of the years involved in the study.

Page 7, line 160-162: Cohort analyses is plural and you only reference a single cohort analysis -13 in references. Should more articles be included in the references?

Material and Methods, Data set: the paragraph causes confusion over what is actually included in the data as its hospital attendances from 2/4/2010 – 31/08/2023 which is not even a full month for April. And then the mention in the final line that only complete calendar years were included. This is not explained further or detailed in the results and leaves me unsure how it was actually done. This needs to be amended to either have the data set from 1/1/2011 to 31/12/2022 which seems the best option or to details all the presentations for the dates 2/4/2010 – 31/08/2023 which is not clearly detailed currently.

Results: no details are provided on the proportion of attendances which required involvement of mental health services or involved injures which I think detracts from the data being presented. No details are provided on any data that was excluded.

Page 25, Lines 469-471 you say: "While a girl’s or women’s alcohol use should have no bearing on how sexual assault is viewed or treated, individuals can place more responsibility on women if they have been drinking alcohol prior to the assault (25)." To date in the article you have made no mention of sexual assault in the results or provided any details on this. I think this statement needs to be removed as it doesn’t seem to follow the flow of the article and there is no details on injury patterns or assaults.

Page 25-26, Lines 483-484 you say: "These concerns included a range of adverse experiences, including sexual abuse, domestic violence and being in social care." This is actual results and should be detailed in the results section rather than discussion. It might be beneficial to have further detail on this in the results to understand the frequency.

Figures: there is inconsistency between the tables in the use of capital letters and this should be consistent in the labels.

Figure 1: This 3 axis graph is harder to read as the 3rd axis (attendances resulting in admission) is offset against the gridlines presented. This chart might be easier to read if different colour gridlines were used to match corresponding sides or the gridlines were removed altogether and a vertical line was placed next to each axis with small horizontal lines corresponding to each point. The two zeros after the decimal point are not needed in the % side and would be easier to read if they were removed.

Figure 2: X axis is labelled attendances rate which appears incorrect and I think should be proportion of alcohol related attendances. The two zeros after the decimal point are not needed and would be easier to read if they were removed.

Reviewer #2: In terms of strength, the study utilizes ED attendance data rather than just admission data, giving a complete picture of alcohol-related harm among this vulnerable population group. The 11 years timeframe allows for meaningful trend analysis. The methodology seems to be rigorous including population-adjusted rates and multivariable models.

However, data taken from a single paediatric ED may not be generalizable to other regions or setting. There's no validation of how accurately alcohol-related attendances were identified in the electronic records which potentiates missed cases if alcohol involvement wasn't properly documented. The author also did not explore if there is potential changes in coding practices over the 11-year period that might influence trends. There was no information of severity of intoxication, types of injuries or other clinical details that might provide more context. However, it may be beyond the scope intended by the author.

Overall, this study provides valuable insights into trends and patterns of alcohol-related ED attendances among young population. The authors work are commendable.

**Do you want your identity to be public for this peer review?** For information about this choice, including consent withdrawal, please see our Privacy Policy

Reviewer #1: **Yes: ** Sheena Durnin

Reviewer #2: **Yes: ** Nurul Huda Ahmad

---

## [Author Response · Author response to Decision Letter 1]

2 Jul 2025

Dear reviewers,

Thank you for reviewing our submission. Please see below for responses to each of your points raised during the review process.

• I have checked the file naming and I cannot see which files do not follow the style requirements – could you please advise?

“We thank the Hugh Greenwood Legacy and the University of Liverpool for supporting this research.”

I have removed the funding information from the acknowledgements section in the manuscript – could you please update the funding statement as given below:

“This research was part of a PhD studentship funded by an external sponsor (Hugh Greenwood Legacy, Alder Children’s Hospital) and the University of Liverpool. The studentship was awarded to AR and SM (as supervisors) and the PhD student was ND. The funders had no involvement in the methods, analysis, or interpretation of the research.

Hugh Greenwood (https://liverpoolhealthpartners.org.uk/the-hugh-greenwood-legacy-for-childrens-health-research/)

University of Liverpool (https://www.liverpool.ac.uk/)”

• As detailed in the Data Availability Statement, the individual level data underlying this study cannot be made publicly available due to restrictions outlined in the Data Management Plan agreed with the NHS provider, this is because sharing would pose a risk to patient confidentiality.

• The form has been updated to include the institutional body to which data requests could be sent.

Comments to the Author – see tracked changes for full details

Page 2, line 30: ONS abbreviation needs to be explained as not previously used or explained.

• “Attendance rates per 100,000 were computed using UK national mid-year population estimates” removed reference to ONS as this is not necessary detail for abstract

Abstract: Page 2-3: needs to include details of the years involved in the study.

• Added, 2011 to 2022 (page 2, line 26)

Page 7, line 160-162: Cohort analyses is plural and you only reference a single cohort analysis -13 in references. Should more articles be included in the references?

• Updated to singular analysis (it was one study) (page 7, line 164)

Material and Methods, Data set: the paragraph causes confusion over what is actually included in the data as its hospital attendances from 2/4/2010 – 31/08/2023 which is not even a full month for April. And then the mention in the final line that only complete calendar years were included. This is not explained further or detailed in the results and leaves me unsure how it was actually done. This needs to be amended to either have the data set from 1/1/2011 to 31/12/2022 which seems the best option or to details all the presentations for the dates 2/4/2010 – 31/08/2023 which is not clearly detailed currently.

• Updated to refer to study period as 1/1/2011 to 31/12/2022 as only these cases were included in the analysis. (Page 9, Line 193). Removed text in page 9 (lines 206 to 207) to avoid confusion).

Results: no details are provided on the proportion of attendances which required involvement of mental health services or involved injures which I think detracts from the data being presented. No details are provided on any data that was excluded.

• Proportion of attendances which required involvement of mental health services was not captured in the data – this is why we used safeguarding concerns as a proxy measure (as safeguarding would include these cases, as well as other concerns e.g. children in social care)

• Similarly, proportion of attendances involving injuries was not complete in the data and included a free text variable which would be difficult to code reliably. Therefore, we used admission to hospital as a proxy for injury severity.

Page 25, Lines 469-471 you say: "While a girl’s or women’s alcohol use should have no bearing on how sexual assault is viewed or treated, individuals can place more responsibility on women if they have been drinking alcohol prior to the assault (25)." To date in the article you have made no mention of sexual assault in the results or provided any details on this. I think this statement needs to be removed as it doesn’t seem to follow the flow of the article and there is no details on injury patterns or assaults.

• We have reworded this paragraph (Page 25, Lines 472 to 479).

Page 25-26, Lines 483-484 you say: "These concerns included a range of adverse experiences, including sexual abuse, domestic violence and being in social care." This is actual results and should be detailed in the results section rather than discussion. It might be beneficial to have further detail on this in the results to understand the frequency.

• This isn’t results – it is how the safeguarding variable was defined (see page 10, line 229 in methods). The safeguarding concerns variable includes cases where children are flagged on the system as being in social care, have had a previous mental health referral, have domestic violence reported etc. It is not possible to split within this variable to see how many fall in each category as it was a binary variable (reported concerns either Yes or No). We did not have access to this sensitive information in more detail, due to data management contracts with the NHS host.

Figure 1: This 3 axis graph is harder to read as the 3rd axis (attendances resulting in admission) is offset against the gridlines presented. This chart might be easier to read if different colour gridlines were used to match corresponding sides or the gridlines were removed altogether and a vertical line was placed next to each axis with small horizontal lines corresponding to each point. The two zeros after the decimal point are not needed in the % side and would be easier to read if they were removed.

• Removed gridlines and decimal points for Fig 1 and added vertical line next to each acis (with small horizontal lines corresponding to each point).

Figure 2: X axis is labelled attendances rate which appears incorrect and I think should be proportion of alcohol related attendances. The two zeros after the decimal point are not needed and would be easier to read if they were removed.

• Updated label and removed decimal places

However, data taken from a single paediatric ED may not be generalizable to other regions or setting. There's no validation of how accurately alcohol-related attendances were identified in the electronic records which potentiates missed cases if alcohol involvement wasn't properly documented. The author also did not explore if there is potential changes in coding practices over the 11-year period that might influence trends. There was no information of severity of intoxication, types of injuries or other clinical details that might provide more context. However, it may be beyond the scope intended by the author.

• The strategy for identifying and defining alcohol related attendances was devised with input from data technicians at Alder Hey and the clinical supervisor (a consultant in paediatric emergency medicine), based on availability of data .

• Severity of intoxication was determined by whether patients were admitted to hospital or discharged. Including other clinical details would be beyond the scope of this study.

• In the limitations section, we had highlight potential for demographic changes within the catchment area over time, diagnostic coding errors, barriers for clinicians to obtain alcohol info in children/young people’s ED settings etc. We also state that findings may not generalise to other regions (line - 541-543). This data comes from Alder Hey Children’s NHS Foundation Trust, one of Europe’s largest and busiest children’s hospitals, providing care for over 450,000 children and young people (from birth to 16 years) each year.

• We have added an outline for future research to build on this evidence

Kind regards,

Nicholas Davies and Abi Rose

---

## [Editor Report · Decision Letter 1]

16 Jul 2025

Identifying and describing alcohol-related paediatric emergency department attendances amongst under 16 year olds including time trends, incidence rates, and sociodemographic factors associated with alcohol-related harm

PONE-D-24-56004R1

Dear Dr. Davies,

We’re pleased to inform you that your manuscript has been judged scientifically suitable for publication and will be formally accepted for publication once it meets all outstanding technical requirements.

Kind regards,

Nik Hisamuddin Nik Ab. Rahman

Academic Editor

PLOS ONE
---

## [Editor Report · Acceptance letter]

PONE-D-24-56004R1

PLOS ONE

Dear Dr. Davies,

I'm pleased to inform you that your manuscript has been deemed suitable for publication in PLOS ONE. Congratulations! Your manuscript is now being handed over to our production team.

Kind regards,

on behalf of

Professor Dr Nik Hisamuddin Nik Ab. Rahman

Academic Editor

PLOS ONE